# siRNA Targeting PDE5A Partially Restores Vascular Damage Due to Type 1 Diabetes in a Streptozotocin-Induced Rat Model

Vanessa Giselle Garcia-Rubio [1], Sandra Edith Cabrera-Becerra [1], Sergio Adrian Ocampo-Ortega [1], Citlali Margarita Blancas-Napoles [1], Vivany Maydel Sierra-Sánchez [1], Rodrigo Romero-Nava [1], Rocío Alejandra Gutiérrez-Rojas [2], Fengyang Huang [3], Enrique Hong [4] and Santiago Villafaña [1,*]

1 Laboratorio de Señalización Intracelular, Sección de Estudios de Posgrado e Investigación, Escuela Superior de Medicina, Instituto Politécnico Nacional, Plan de San Luis y Salvador Díaz Mirón, Ciudad de México 11340, Mexico
2 Escuela Nacional de Ciencias Biológicas, Instituto Politécnico Nacional, Ciudad de México 11340, Mexico
3 Departamento de Farmacología y Toxicología, "Hospital Infantil de México Federico Gómez" (HIMFG), Ciudad de México 06720, Mexico
4 Departamento de Neurofarmacobiología, Centro de Investigación y de Estudios Avanzados, Ciudad de México 07360, Mexico
* Correspondence: svillafana@ipn.mx; Tel.: +52-55-57-29-63-00 (ext. 62824)

**Abstract:** Diabetes mellitus is a metabolic disease that can produce different alterations such as endothelial dysfunction, which is defined as a decrease in the vasodilator responses of the mechanisms involved such as the nitric oxide (NO) pathway. The overexpression of PDE5A has been reported in diabetes, which causes an increase in the hydrolysis of cGMP and a decrease in the NO pathway. For this reason, the aim of this study was to evaluate whether siRNAs targeting PDE5A can reduce the endothelial dysfunction associated with diabetes. We used male Wistar rats (200–250 g) that were administered streptozotocin (STZ) (60 mg/kg i.p) to induce diabetes. Two weeks after STZ administration, the siRNAs or vehicle were administered and then, at 4 weeks, dose–response curves to acetylcholine were performed and PDE5A mRNA levels were measured by RT-PCR. siRNAs were designed by the bioinformatic analysis of human–rat FASTA sequences and synthesised in the Mermade-8 equipment. Our results showed that 4 weeks of diabetes produces a decrease in the vasodilator responses to acetylcholine and an increase in the expression of PDE5A mRNA, while the administration of siRNAs partially restores the vasodilator response and decreases PDE5A expression. We conclude that the administration of siRNAs targeting PDE5A partially reverts the endothelial impairment associated with diabetes.

**Keywords:** diabetes; PDE5A-siRNA; endothelial dysfunction; PDE5A mRNA; vasodilation

## 1. Introduction

Diabetes mellitus is a group of metabolic diseases characterised by hyperglycaemia resulting from defects in insulin secretion, insulin action or both [1]. It has been estimated that the global prevalence of type 1 diabetes is 5.9 per 10,000 people, which has been increasing over the years [2]. In 2021, diabetes caused 1.7 million deaths worldwide [3]; the burden of type 1 diabetes in 2021 was vast and expected to increase rapidly, especially in resource-limited countries [4]. It has been reported that chronic exposure to hyperglycaemia produces different complications, which can be macrovascular, such as coronary artery, cerebrovascular and peripheral vascular diseases or microvascular, for example, retinopathy, nephropathy, neuropathy, etc. [5]. The repercussions of macrovascular complications are that they increase the risk of cardiovascular morbidity and mortality by 2–4 times, while, on the other hand, microvascular complications markedly affect the quality of life of diabetic people and cause disability [6]. The first event that gives rise to the generation of these complications is endothelial dysfunction [7]. Endothelial dysfunction is commonly defined

as changes that lead to a reduction in the responses that generate endothelium-dependent vasodilation, for example, acetylcholine or flow-mediated vasodilatation [8]; therefore, we cannot exclude the possibility that acting on endothelial dysfunction may delay the onset of diabetes mellitus complications and reduce cardiovascular mortality and morbidity [9]. Interestingly, the current treatment against endothelial dysfunction associated with type 1 diabetes mellitus is based on lifestyle interventions, glycaemic control and antioxidant therapy [10,11]; however, the results are controversial, and no significant benefits were seen in some cases [12–14], but research has shown some benefits of blueberry consumption [15,16], which suggests that is necessary to explore new mechanisms. Recent studies have reported that hyperglycaemia leads to the overexpression of type 5 phosphodiesterase (PDE5A) [17] and that its inhibition improves brachial arterial flow [18] and improves vasodilation [19]; this is because this enzyme hydrolyses cGMP to inactive GMP [20] and limits the activity of the NO–PKG pathway [21], promoting endothelial dysfunction.

The human PDE5A gene is located on chromosome 4q26, containing 23 exons of approximately 100 kb; the first three exons are alternative exons that encode three isoforms: PDE5A1, PDE5A2 and PDE5A3 [22], of which the PDE5A1 and PDE5A2 isoforms have also been found in dogs and rats [23,24]. These isoforms are expressed widely in various tissues, such as vascular smooth muscle cells, penile corpus cavernosum, lung, kidney, brain, cardiac myocytes, gastrointestinal tissue and platelets [25,26], while PDE5A3 has only been found in humans and is considered smooth muscle-specific [23].

Currently, some PDE5A inhibitors (PDE5Ai) have been used, showing that they can enhance the vasodilation mediated through the NO/cGMP pathway [27]; for this reason, they have been applied as treatments for erectile dysfunction, pulmonary hypertension and heart failure, although recent preclinical studies suggest that PDE5Ai can be useful for diabetic neuropathy [28]. However, none of the PDE5Ai are genuinely selective for PDE5A, and most of the adverse reactions are due to its non-selective interaction with other PDE isoenzymes; for example, blue colour vision, back pain, myalgias, dyspepsia, headache, flushing, hypotension and dizziness may be primarily due to interactions between PDE6 and PDE11 [29–31].

Recently, siRNAs have been used as a treatment for different pathologies; siRNAs are short synthetic RNA duplexes [32] that are able to repress the expression of specific genes using the complementarity between sequences [33]. These double strands of RNA were shown to have greater interference efficiency than single RNA sequences [34]; the process by which this silencing takes place is known as post-transcriptional gene silencing [35].

For this reason, we propose the use of small interfering RNAs (siRNAs) directed to PDE5A to reduce cardiovascular damage associated with diabetes. Our research question is whether the use of siRNA could reduce endothelial dysfunction associated with diabetes and our hypothesis is that the administration of siRNA will improve the endothelial dysfunction associated with diabetes.

## 2. Materials and Methods

### 2.1. Animals

Male Wistar rats weighing 200–250 g were used; the animals were kept in an isolated room at room temperature, with a light/dark cycle of 12 h, a temperature of 25 °C ± 2 and access to water and food ad libitum. They were randomly divided into 4 groups ($n = 6$): (1) control group (citrate buffer); (2) diabetic group; (3) diabetic group + transfection vehicle; and (4) diabetic group + siRNA. The experimental procedures were conducted following the regulations proved by our Institutional Committee for the Care and Use of Laboratory Animals (ESM-CICUAL-ADM-01/27-09-2019) and the Official Mexican Standard NOM-062-ZOO-1999 Technical specifications for the production, care and use of laboratory animals. The minimum number of animals was used per group according to the 3Rs (replacement, reduction and refinement); an $\alpha = 0.05$, $\beta = 0.2$, population variance = 6.342 and allowable difference = 3 obtained a $n = 6$ with the software sample size calculator: one sample mean [36], which was approved by the animal use committee.

Animals were randomly placed into groups; random numbers were generated using the standard = RAND() function in Microsoft Excel.

### 2.2. Model of Type 1 Diabetes Mellitus

The rats were injected intraperitoneally with streptozotocin (STZ) (Sigma-Aldrich, St. Louis, MO, USA) at a dose of 60 mg/kg dissolved in a citrate buffer (0.1 M pH 4.5), while the control rats were administered citrate buffer [37].

### 2.3. Measurement of Glucose, Heart Rate, Systolic and Diastolic Blood Pressure

After one week of STZ treatment, blood glucose levels were determined using a glucometer (Accu-Check Performa, Mannheim, Germany); blood samples were obtained from the rat tail by a small cut and glucose levels greater than $\geq$300 mg/dL were considered type 1 diabetes [37]. Heart rate and systolic and diastolic pressure were measured using the tail-cuff method (IITC Life Science Inc., Woodland Hills, CA, USA) [38]. Prior to measurements, the rats underwent training and were placed in a stock according to their size (Rodent Restrainer MLA5022), with their tails placed in a B60-7/16″ tail cuff, at a temperature of 32 °C for 20 min for 5 consecutive days. Hypertension was considered when the values of systolic arterial pressure were above 140 mmHg for systolic blood pressure and 90 mmHg for diastolic blood pressure.

### 2.4. Dose–Response Curves to Acetylcholine

After four weeks of citrate buffer or STZ administration, dose–response curves for acetylcholine were determined. Twenty-four hours before the dose–response curves to acetylcholine, rats were reserpinised with 5 mg/kg i.p. (Sigma-Aldrich, St. Louis, MO, USA) and dissolved in dimethyl sulphoxide (DMSO) (Sigma-Aldrich, St. Louis, MO, USA) to perform a chemical sympathectomy. After that, the rats were anaesthetised with sodium pentobarbital 50 mg/kg i.p. in a 1:1 dilution with 0.9% saline solution. To provide assisted breathing, an endotracheal tube was inserted through a tracheostomy and connected to the Harvard Apparatus Rodent Respirator Model 680; ventilatory parameters were adjusted according to animal weight. The external carotid artery was cannulated and connected to a Statham Gould transducer coupled to a Grass model 7D polygraph with a data acquisition system (Polyview) to record changes in blood pressure. The femoral vein was cannulated for drug administration; to evaluate the vasodilator responses, an acetylcholine hydrochloride dose of 0.00001 to 10 μg/kg i.v. was used (Sigma-Aldrich, St. Louis, MO, USA).

### 2.5. Evaluation of the PDE5A Expression

For evaluation of the relative expression of the PDE5A mRNA, four weeks after STZ administration, an overdose of pentobarbital at 80 mg/kg i.p. was applied to euthanise the rats; the aorta was carefully removed and cleaned of blood and connective tissue and subsequently placed in 1 mL of TRIZOL reagent (Roche Diagnostics, Indianapolis, IN, USA) to extract total RNA according to the manufacturer's instructions. Concentration and purity were determined by measuring the optical densities at 260/280 nm with the Nanophotometer Pearl (IMPLEN, München, Germany) [39]. Total RNA was diluted to 1 μg/mL with nuclease-free water; the cDNA synthesis was conducted using the Transcriptor First Strand cDNA Synthesis Kit (Roche Diagnostics, Indianapolis, IN, USA). The following primers were used to measure PDE5A mRNA: CCATGTGGTGT-GAACAACT (sense) and GGTCGAAGTGATGGTGCTC (antisense) and HRPT mRNA CAATCAAGACGTTCTTTCCAGTT (sense) and GCTCCATTCCTATGACTGTAGATTTT (antisense). RT-PCR without cDNA was considered the negative control [40]. The expression levels of PDE5A mRNA were determined using the comparative method $2^{-\Delta\Delta Ct}$, normalising the expression with HPRT [41]. This method is the most sensible assay to demonstrate the siRNA efficiency because it avoids problems with the half-life of the proteins that could mask the silencing efficiency [42].

*2.6. PDE5A siRNA Design and Synthesis*

The PDE5A FASTA sequences of rats and humans were searched in the GenBank database and placed in the Wizard v3.1 software to find regions that are susceptible to silencing. Regions with a length of 21–25 nucleotides and guanine–cytosine (GC) content of 35–55%, were selected; the following regions were excluded: TTTT, GGTGTT, GGGGGG, CCCCCC, GGCCGGC, GTCCTTCAA, TGTGT and CTGAATT, because it has been reported that they can induce the activation of TLR 7/8 receptors [43]. An alignment of the FASTA sequences of the human and rat PDE5A mRNA was performed using the Clustal Omega program, where it was verified that the selected sequences share 100% similarity [44]. The hybridisation site was identified in the secondary structure of minimum energy (MFE), using the RNA-fold webserver software; the most accessible regions to hybridise were identified and then the regions were blasted in the NCBI nucleotide blast software to identify the likely off-target sequences. Any sequences that shared >78% similarity and more than 15 identical nucleotides were excluded to avoid nonspecific hybridisation. The selected sequences were synthesised using the Mermade-8 equipment through the following steps: deblocking, coupling, capping and oxidation. Subsequently, cleavage with 30% ammonium hydroxide was carried out along with RNA 2′deprotection with a solution of 1M triethylamine trihydrofluoride; the sequences were desalted and hybridised in a HEPES buffer solution.

*2.7. Functional Evaluation of the Effect of siRNA Directed to PDE5A*

Two weeks after the administration of STZ ($t_2$), the rats were anaesthetised with pentobarbital 30 mg/kg i.p. and 5 μg/kg of the synthesised siRNAs were administered in the jugular vein. Subsequently, two weeks after the administration of siRNAs ($t_4$), cardiovascular function was evaluated using dose–response curves to acetylcholine at increasing doses along with the relative expression of PDE5A mRNA in aortic tissue.

*2.8. Statistical Analysis*

Data were expressed as the mean ± SEM. A statistical comparison between control (non-diabetic rats) and diabetic rats was performed using the unpaired student *t*-test; for more than two groups, statistical differences were analysed using one-way ANOVA followed by Bonferroni's post hoc test. Differences were considered statistically significant when $p < 0.05$ vs. control.

**3. Results**

After four weeks of STZ administration, our results showed a significant increase in blood glucose levels and a decrease in weight and heart rate compared to the control group; on the other hand, we did not observe any changes in blood pressure (Table 1). With respect to dose–response curves, a significant decrease in vasodilation was induced by acetylcholine in diabetic rats compared to non-diabetic rats at a dose of 0.001–10 μg/kg (Figure 1). The transcriptional analysis of PDE5A expression showed a significant increase in mRNA levels in the aorta of diabetic rats in comparison with non-diabetic rats (Figure 2).

**Table 1.** Physiological parameters.

|  | Control | Diabetes | Diabetes Vehicle | Diabetes siRNA |
|---|---|---|---|---|
| Weight (g) | 384.40 ± 6.43 | 223.40 ± 5.96 * | 190.67 ± 9.74 | 276.67 ± 14.53 |
| Glucose (mg/dL) | 113.56 ± 1.33 | 529.67 ± 18.96 * | 517.78 ± 12.22 | 410.89 ± 17.42 |
| HR (beats/min) | 414.73 ± 10.47 | 330.18 ± 4.24 * | 344.44 ± 16.64 * | 355.25 ± 8.08 |
| SAP (mm Hg) | 117.63 ± 1.02 | 118.09 ± 1.15 | 118.00 ± 0.24 | 118.33 ± 0.29 |
| DAP (mm Hg) | 77.40 ± 1.12 | 85.46 ± 1.13 | 81.18 ± 18 | 78.09 ± 2.31 |

Heart rate (HR), systolic arterial pressure (PAS), diastolic arterial pressure (DAP). Values are expressed as mean ± standard error of the mean (SEM). * $p \leq 0.05$ vs. controls (non-diabetic rats).

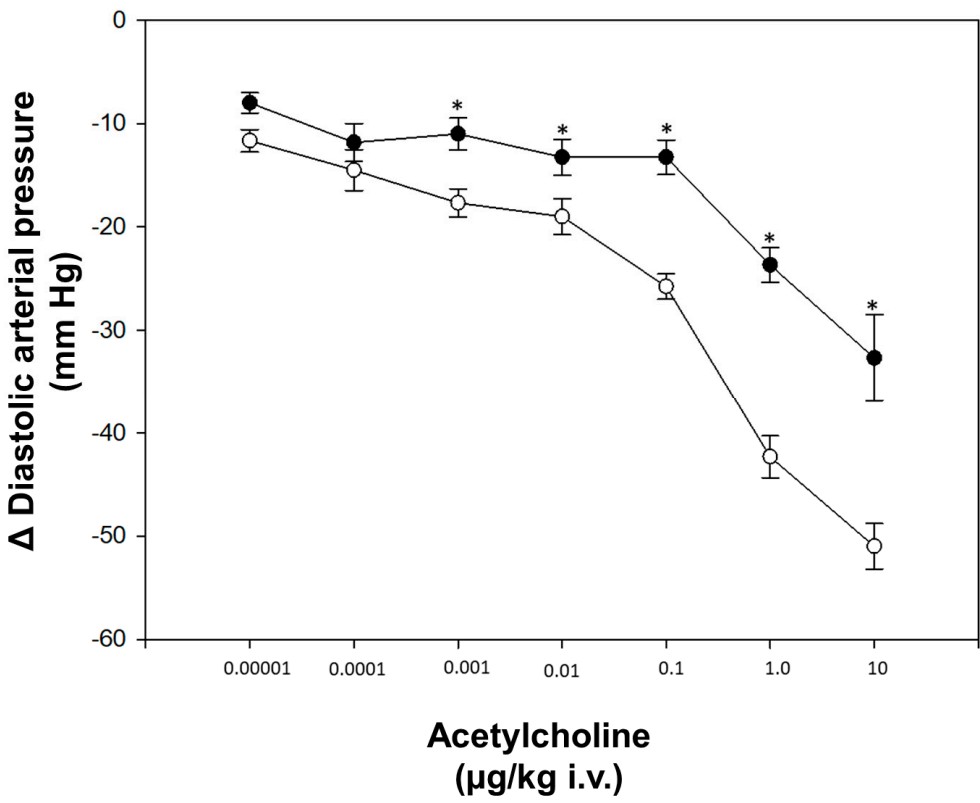

**Figure 1.** Dose-response curve. (○) Control (non-diabetic rats), (●) diabetic rats. Values express the mean ± standard error. * $p < 0.05$ vs. control group.

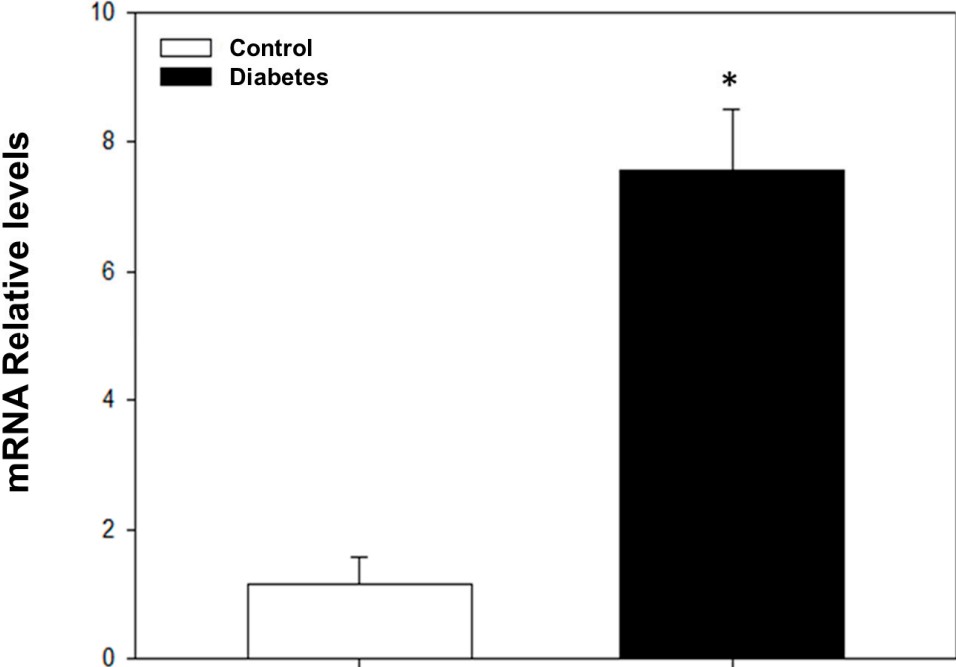

**Figure 2.** mRNA relative levels determined by RT-PCR for PDE5A. The values express the mean ± standard error of the PDE5A mRNA relative expression levels HRPT. * $p < 0.05$ vs. control group.

Regarding the siRNA design, the bioinformatic analysis carried out using the siRNA Wizard software showed that there were conserved regions in the FASTA sequence of three human variants and rats that were susceptible to silencing (Table 2); we selected two sequences

(a) 5′-GCAGAACTTCCAGATGAAACA-3′ and (b) 5′-GCCTCTCCATTGAAGAATATA-3′, which are in the coding sequences (CDS) in both species with predominantly hairpin loop structures (Table 3). Secondary structure analysis showed regions that are susceptible to hybridisation (Figure 3) and with suitable physicochemical parameters (Tables 4 and 5).

**Table 2.** PDE5A mRNA data in the siRNA hybridisation regions.

**siRNA-1. Sequence: GCAGAACTTCCAGATGAAACA**

| | | |
|---|---|---|
| NM_133584.1 | TGTCTGATCTGGAAACAGCGCTGTGTACAATTCGGATGTTCACTGACCTCAACCTTGTGC | 2104 |
| NM_001083.4 | TGTCTGATCTGGAAACAGCACTGTGTACAATTCGGATGTTTACTGACCTCAACCTTGTGC | 1889 |
| NM_033430.3 | TGTCTGATCTGGAAACAGCACTGTGTACAATTCGGATGTTTACTGACCTCAACCTTGTGC | 2192 |
| NM_033437.3 | TGTCTGATCTGGAAACAGCACTGTGTACAATTCGGATGTTTACTGACCTCAACCTTGTGC | 1694 |
| NM_133584.1 | AGAACTTCCAGATGAAACACGAGGTTCTTTGCCGATGGATTTTGAGTGTCAAGAGAATT | 2164 |
| NM_001083.4 | AGAACTTCCAGATGAAACATGAGGTTCTTTGCAGATGGATTTTAAGTGTTAAGAAGAATT | 1949 |
| NM_033430.3 | AGAACTTCCAGATGAAACATGAGGTTCTTTGCAGATGGATTTTAAGTGTTAAGAAGAATT | 2252 |
| NM_033437.3 | AGAACTTCCAGATGAAACATGAGGTTCTTTGCAGATGGATTTTAAGTGTTAAGAAGAATT | 1754 |

**siRNA-2. Sequence: GCCTCTCCATTGAAGAATATA**

| | | |
|---|---|---|
| NM_133584.1 | TCGACCAGTGCTTGATGGTTCTAAACAGCCCAGGCAACCAGATCCTCATGGCCTCTCCA | 2464 |
| NM_001083.4 | TTGACCAGTGCCTGATGATTCTTAATAGTCCAGGCAATCAGATTCTCAGTGGCCTCTCCA | 2249 |
| NM_033430.3 | TTGACCAGTGCCTGATGATTCTTAATAGTCCAGGCAATCAGATTCTCAGTGGCCTCTCCA | 2552 |
| NM_033437.3 | TTGACCAGTGCCTGATGATTCTTAATAGTCCAGGCAATCAGATTCTCAGTGGCCTCTCCA | 2054 |
| NM_133584.1 | TTGAAGAATATAAGACCACATTGAAAATAATCAAGCAAGCAATTTTAGCCACTGACTAG | 2524 |
| NM_001083.4 | TTGAAGAATATAAGACCACGTTGAAAATAATCAAGCAAGCTATTTTAGCTACAGACTAG | 2309 |
| NM_033430.3 | TTGAAGAATATAAGACCACGTTGAAAATAATCAAGCAAGCTATTTTAGCTACAGACTAG | 2612 |
| NM_033437.3 | TTGAAGAATATAAGACCACGTTGAAAATAATCAAGCAAGCTATTTTAGCTACAGACTAG | 2114 |
| NM_133584.1 | *Rattus norvegicus* phosphodiesterase 5A (Pde5a), mRNA | |
| NM_001083.4 | *Homo sapiens* phosphodiesterase 5A (PDE5A), transcript variant 1, mRNA | |
| NM_033430.3 | *Homo sapiens* phosphodiesterase 5A (PDE5A), transcript variant 2, mRNA | |
| NM_033437.3 | *Homo sapiens* phosphodiesterase 5A (PDE5A), transcript variant 3, mRNA | |

**Table 3.** Coordinates, types of secondary structures and silence region of PDE5a mRNA in the hybridisation regions of siRNA-1.

| Species | Coordinate | Secondary Structure | Silence Region |
|---|---|---|---|
| *Rattus norvegicus* (NM_133584.1) | 2103–2124 | Stem-hairpin loop | CDS (exon 11) |
| *Homo sapiens Variant 1* (NM_001083.4) | 1888–1909 | Hairpin loop | CDS (exon 11) |
| *Homo sapiens* Variant 2 (NM_033430.3) | 2191–2212 | Stack-hairpin loop | CDS (exon 11) |
| *Homo sapiens* Variant 3 (NM_033437.3) | 1693–1714 | Stack-hairpin loop | CDS (exon 10) |

CDS: coding sequences.

**Table 4.** Coordinates, types of secondary structures and silence region of PDE5a mRNA in the hybridisation regions of siRNA-2.

| Species | Coordinate | Secondary Structure | Silence Region |
|---|---|---|---|
| *Rattus norvegicus* (NM_133584.1) | 2456–2477 | Loop-stack-loop | CDS (exon 15) |
| *Homo sapiens Variant* 1 (NM_001083.4) | 2241–2262 | Hairpin loop-bulge | CDS (exon 16) |
| *Homo sapiens* Variant 2 (NM_033430.3) | 2544–2565 | Hairpin loop | CDS (exon 15) |
| *Homo sapiens* Variant 3 (NM_033437.3) | 2046–2067 | Stack-hairpin loop | CDS (exon 14) |

CDS: coding sequences.

Our findings after siRNA administration show a significant decrease in the relative expression of PDE5A mRNA and an increase in the vasodilatation induced by acetylcholine (Figures 4 and 5). Moreover, diabetic rats treated with PDE5A siRNA showed a decrease in blood glucose levels and weight gain in comparison to non-treated diabetic rats (Table 1).

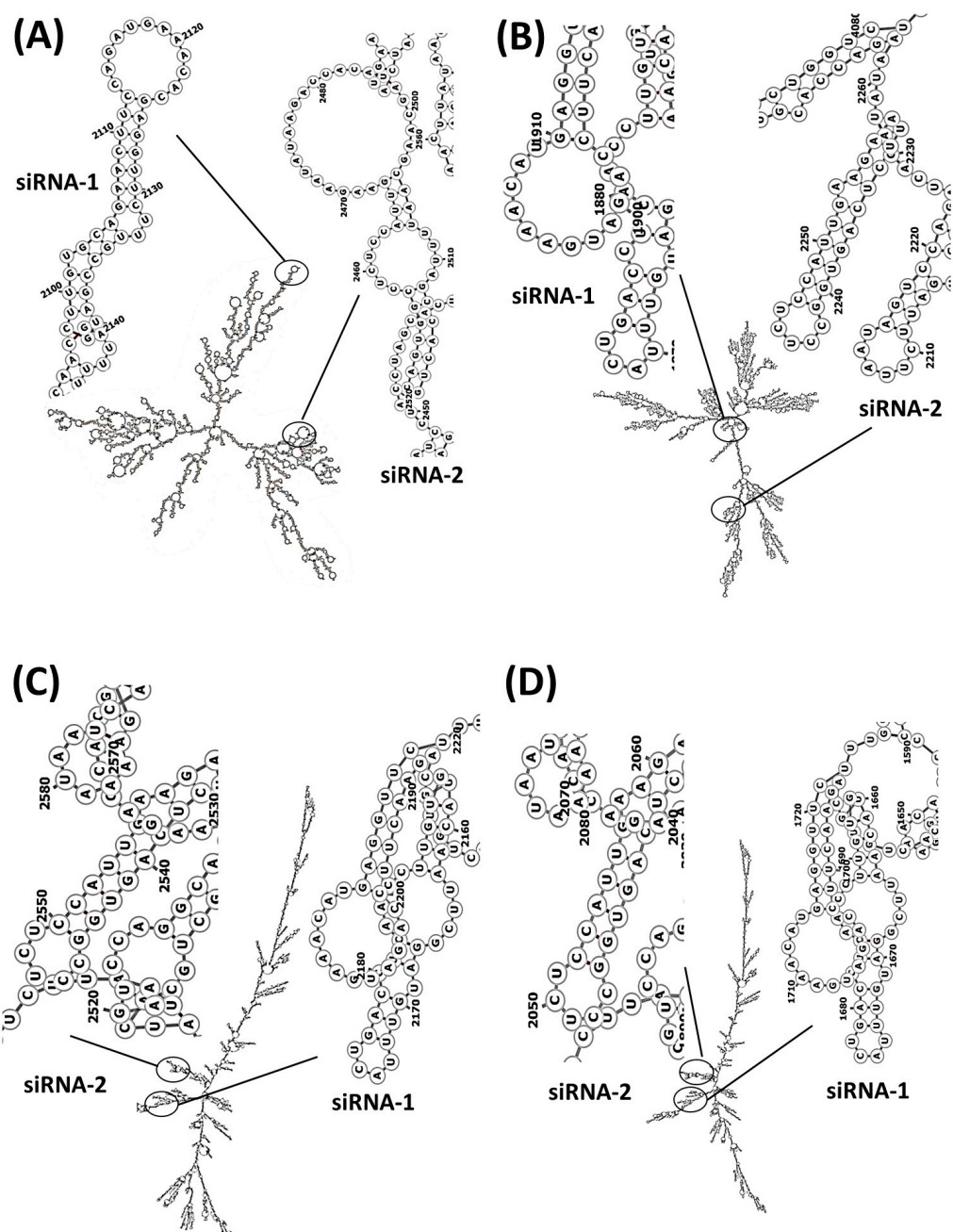

**Figure 3.** PDE5A mRNA secondary structures and hybridisation regions of siRNA sequences directed to PDE5A. (**A**) Rattus norvegicus, (**B**) *Homo sapiens* variant 1, (**C**) *Homo sapiens* variant 2 and (**D**) *Homo sapiens* variant 3. The formation of secondary structures was carried out by calculating the centroid of the RNA sequences.

**Table 5.** Physicochemical parameters.

| Sequence | TM (°C) | MW (g/mol) |
|---|---|---|
| PDE5A18 (G) | 53.1 | 6432.3 |
| PDE5A23 (G) | 49.3 | 6389.2 |
| PDE5A18 (P) | 53.1 | 6418.2 |
| PDE5A23 (P) | 49.3 | 6460.3 |

Abbreviatures: guide strand (G), passenger strand (P), temperature melting (TM), molecular weight (MW).



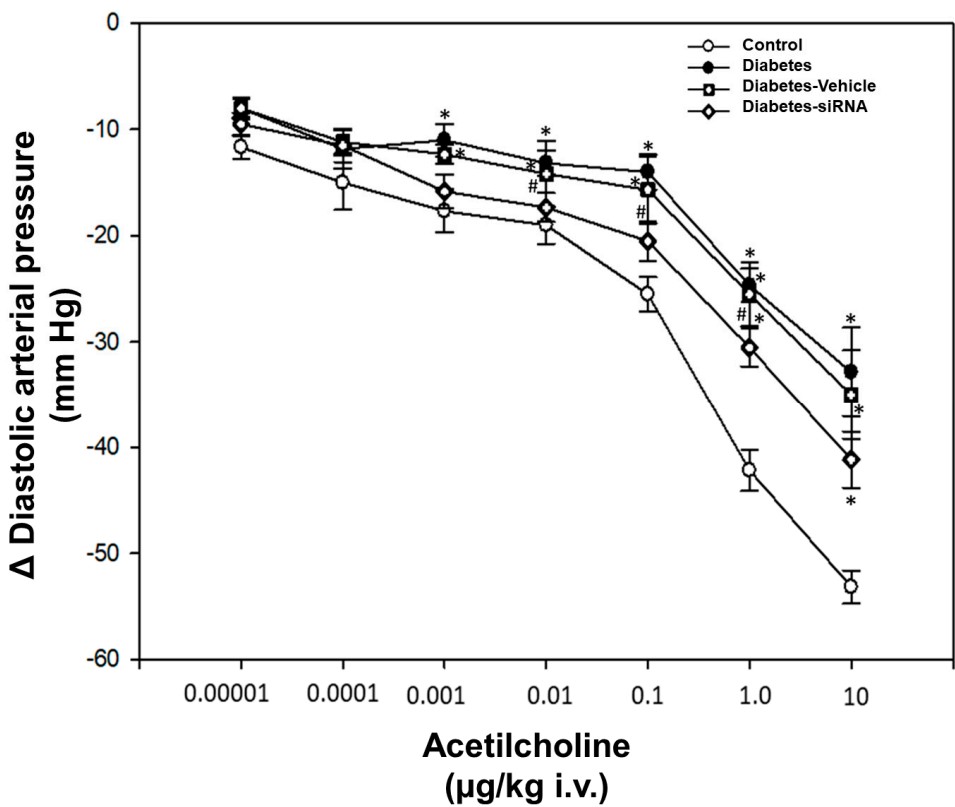

**Figure 4.** Dose-response curve. Values express the mean ± standard error. * $p < 0.05$ vs. control group and # $p < 0.05$ vs. diabetes-vehicle group.

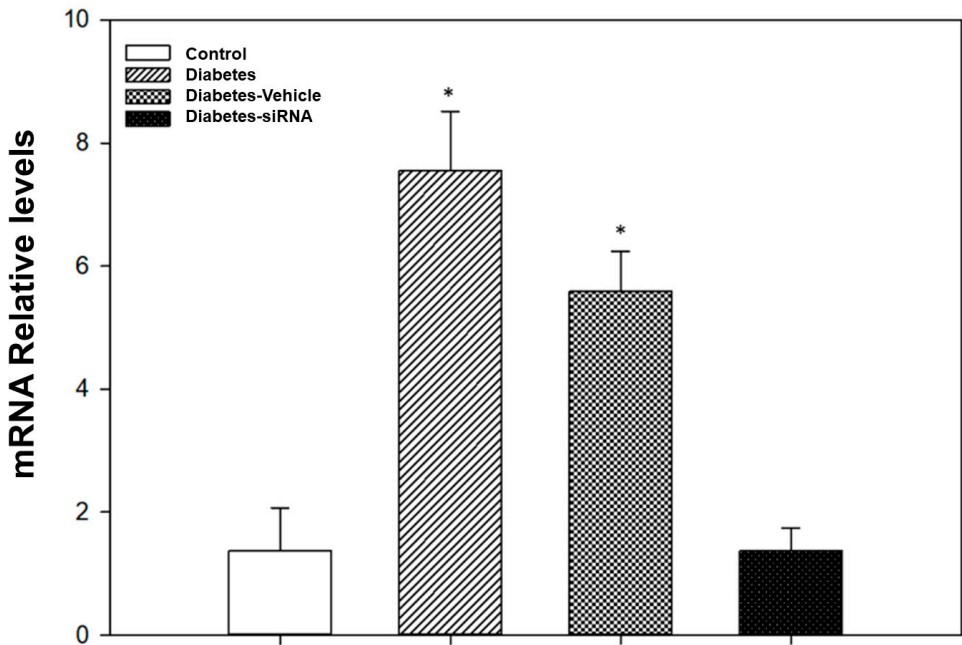

**Figure 5.** Evaluation of gene knockdown efficiency by PDE5A siRNA. * $p < 0.05$ vs. control group.

## 4. Discussion

Our results showed that the STZ administration produced an increase in the glucose levels, which is consistent with previous studies [45,46]; it has been suggested that this is due to the ability of STZ to enter β cells through GLUT2 transporters and induce DNA methylation within the cell as well as increase NADPH levels and free radicals, which causes

extensive beta-cell necrosis, leading to the development of diabetes [47,48]. Interestingly, significant weight loss was also observed in this group, which is likely to be the result of skeletal muscle proteolysis due to the destruction of pancreatic β cells and a lack of insulin production [49,50]. Moreover, a significant decrease in heart rate was seen in diabetic rats, which is consistent with other studies [51,52]. This effect has been associated with cardiac autonomic dysfunction due to neuropathy caused by high glucose levels [53,54]. On the other hand, we did not observe any changes in blood pressure in rats treated with streptozotocin, which is consistent with the results obtained by previous studies [55,56]; however, other groups have shown changes in blood pressure [57,58], possibly due to the strain and age of the animals.

On the other hand, our results in the dose–response curves for acetylcholine show that there is an impaired vasodilatation response in diabetic rats, as reported in earlier studies [59–61]. Endothelial dysfunction is characterised by a reduction in nitric oxide generation, or an unbalanced distribution of the relaxing and contracting pressures produced by endothelial cells [62] and has been defined as an attenuated response of the blood vessels to endothelium-dependent vasodilators such as acetylcholine (ACh) and bradykinin [63]. This has been widely reported with diabetes mellitus; the principal mechanism described has been the minor bioavailability of nitric oxide. Chronic hyperglycaemia increases protein kinase C (PKC) activity [64], which leads to the release of reactive oxygen species [65] and causes an uncoupling of endothelial nitric oxide synthase (eNOS) through superoxide anions [66] and peroxynitrite formation [67], thus reducing the synthesis of nitric oxide [68].

Regarding the transcriptional overexpression of PDE5A due to diabetes observed in this study, some studies have reported increased PDE5 expression due to diabetes [69–71]; this variation in PDE5 expression in diabetes is possibly due to activation by the transcription factors AP-2 and SP-1 [72], because there is evidence that the transcription factor AP-2 participates in diabetes, increasing insulin resistance by reducing the expression of the IRS-1 gene [73]. In the case of SP-1, diabetes has been found to increase its interaction with promoter regions [74]. Also, we cannot rule out oxidative stress, which has been found to regulate the expression of PDE5 in heart failure [75] and especially anion superoxide from NADPH oxidase, because this molecule has been found to stimulate the synthesis of PDE5 through Rho signalling [76]. Another important molecule is the transcription factor NKκB, which binds to promoter regions of PDE5 resulting in upregulation; it is suggested that this is the mechanism that LPS uses to increase the expression of PDE5 [77].

PDE5A transcript analysis showed that there were regions susceptible to silencing which were inside the CDS regions; these regions have been described to produce effective silencing [78]. On the other hand, the secondary structure showed that selected regions primarily contain stem, loops and stem + loops, which have been reported to increase silencing efficiency [79]. The administration of siRNA produced a decrease in the expression of PDE5A, which agrees with previous studies of siRNA where it was shown that siRNA decreases expression through the degradation of messenger RNA [80,81]. This silencing caused an improvement in the vasodilator response, suggesting that endothelial dysfunction in diabetes is partly caused by an alteration in the nitric oxide pathway and especially by the degradation of cGMP; this is because PDE5 inhibition has been reported to enhance NO-induced vasodilator responses by increasing cGMP levels [82,83]. For this reason, we cannot exclude the possibility that the mechanisms of action are shared with respect to the nitric oxide pathway; also, some studies showed that a decrease in mRNA with the siRNA of PDE5A causes a decrease in the protein, which is related to an increase in cGMP, suggesting a mechanism by which the siRNA targets PDE5 in the nitric oxide pathway [81,84].

The development of drugs based on siRNAs offers another way to reduce the development of cardiovascular damage associated with diabetes due to its high selectivity [85], but we can discard some adverse reactions that are present in all siRNAs [86]. However, the unwanted effects present in siRNAs are associated with the length of the sequences where

27 mers are more toxic than 21 mers and the vehicle and duration of the treatments can produce an immune response [87].

The decrease in PDE5 expression due to the vehicle could be due to the cytotoxic effects that different transfection agents usually induce [87,88]; however, the effect of siRNAs is significantly different from the effects of the vehicle, suggesting that the effect is due to silencing, as demonstrated in the determinations of mRNA.

## 5. Conclusions

siRNAs targeting PDE5A partially reduce diabetes-associated endothelial dysfunction due to the degradation of PDE5 mRNA; this effect is seen because siRNAs efficiently hybridise coding regions. We cannot exclude possible future use in the treatment of cardiovascular diseases.

**Author Contributions:** Conceptualisation, V.G.G.-R. and S.V.; Methodology, V.G.G.-R., S.E.C.-B., S.A.O.-O., C.M.B.-N., V.M.S.-S. and S.V.; Formal Analysis, V.G.G.-R., S.E.C.-B., S.A.O.-O., C.M.B.-N., V.M.S.-S., R.A.G.-R., R.R.-N. and S.V.; Investigation, V.G.G.-R., S.E.C.-B., S.A.O.-O., C.M.B.-N., V.M.S.-S., F.H., E.H. and S.V.; Writing—Original Draft Preparation, V.G.G.-R., S.E.C.-B., S.A.O.-O., C.M.B.-N., V.M.S.-S., R.A.G.-R., R.R.-N., F.H., E.H. and S.V.; Writing—Review and Editing, V.G.G.-R., S.E.C.-B., S.A.O.-O., C.M.B.-N., V.M.S.-S., R.A.G.-R., R.R.-N., F.H., E.H. and S.V.; Funding Acquisition, S.V. All authors have read and agreed to the published version of the manuscript.

**Funding:** This research protocol was supported by grants from Secretaría de Investigación y Posgrado del Instituto Politécnico Nacional (Proyectos Insignia IPN-2015 and SIP-IPN: 20231313) and García-Rubio VG received a master fellowship from CONACYT (1032702).

**Institutional Review Board Statement:** The experimental procedures were conducted following the regulations proved by our Institutional Committee for the Care and Use of Laboratory Animals (reference: ESM-CICUAL-ADM-01/27-09-2019) and the Official Mexican Standard NOM-062-ZOO-1999 Technical specifications for the production, care and use of laboratory animals.

**Informed Consent Statement:** Not applicable.

**Data Availability Statement:** All the relevant data found in the study are available in the article.

**Conflicts of Interest:** The authors declare that there are no conflict of interest. The authors alone are responsible for the content and writing of the article.

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
