# Peer review of "siRNA Targeting PDE5A Partially Restores Vascular Damage Due to Type 1 Diabetes in a Streptozotocin-Induced Rat Model"

_scipharm, doi:10.3390/scipharm91040052_

Round 1
Reviewer 1 Report
Comments and Suggestions for Authors
The manuscript submitted for publication by Garcia-Rubio et al., titled: "siRNA targeting PDE5 partially restores vascular damage due to diabetes type 1 in a streptozotocin-induced rat model" is an interesting in vivo study aiming to investigate the effect go siRNA targeting on PDE5 specifically in the context of vascular health/damage in a type 1 diabetes induced rat model.
This is an interesting topic on the mechanistic level with significant potential for clinical and therapeutic applications given that diabetic complications particularly as manifested at the vascular system are the ones that typically induce significant secondary pathologies that lead to organ failures and finally death, when diabetes is not adequately managed. Thus any approach that can lead to improved vascular health can be a helpful aid towards better outcomes and improved quality of life and longevity among people with diabetes.
The reviewer would like to offer a few points for consideration by the authors below for the improvement of the manuscript:
1. The introduction section could benefit from some brief information on the statistics, prevalence and burden of disease regarding diabetes.
2. Consider including an explicit research question with the rationale (reframing somewhat the information already presented) and include a hypothesis statement at the end of the introduction section.
3. How was the number of animals determined (power calculation)?
4. Were the animals blocked by weight before random assignments to their respective groups?
5. How was the dosing and protocol for diabetes induction selected (include citation).
6. The materials and methods section would benefit from more extensive referencing as per the methods and rationales used to select the procedures mentioned.
7. It is important to discuss briefly any potential (predicted) side effects especially in thinking long-term effects of such an approach like the proposed one if it were to be conducted in an extending time line experiment and what those implications may be at the clinical level.
8. Similar improvements in vasculature have been reported through dietary interventions especially with rich in antioxidant fruits. It would be worth including a brief discussion in this regard to link the topic to the clinical application. Below are a couple of publications that may be found useful in this regard:
Vendrame S, Daugherty A, Kristo AS, Riso P, Klimis-Zacas D. Wild blueberry (Vaccinium angustifolium) consumption improves inflammatory status in the obese Zucker rat model of the metabolic syndrome. J Nutr Biochem. 2013 Aug;24(8):1508-12. doi: 10.1016/j.jnutbio.2012.12.010.
Vendrame S, Kristo AS, Schuschke DA, Klimis-Zacas D. Wild blueberry consumption affects aortic vascular function in the obese Zucker rat. Appl Physiol Nutr Metab. 2014 Feb;39(2):255-61. doi: 10.1139/apnm-2013-0249.
Comments on the Quality of English LanguageEnglish language is OK overall, but a read through by a native English speaker would improve syntax, grammar and optimize flow of the manuscript's narrative.
Author Response
Dear reviewer;
We made all the suggested changes and tried to answer all your questions.
1.-The introduction section could benefit from some brief information on the statistics, prevalence and burden of disease regarding diabetes.
That is a good recommendation, we added information about it.
Line 40-42. In 2021, diabetes has caused 1.7 million deaths worldwide [3]; the burden of type 1 diabetes in 2021 was vast and expected to increase rapidly, especially in resource-limited countries [4]
2.- Consider including an explicit research question with the rationale (reframing somewhat the information already presented) and include a hypothesis statement at the end of the introduction section.
That is a good recommendation; for this reason, we added the question and hypothesis at the end of the introduction.
Line 90-95. Our research question is whether the use of siRNA could reduce endothelial dysfunction associated with diabetes and our hypothesis is that the administration of siRNA will improve the endothelial dysfunction associated with diabetes.
3.- How was the number of animals determined (power calculation)?
This is an interesting question; we used the software Sample Size Calculator: one sample mean with a α=0.05, β=0.2, population variance=6.342 and allowable difference=3 and obtained a n=6. We added a sentence.
Line 106-109. The minimum number of animals was used per group according to the 3Rs (Replacement, Reduction and Refinement); an α=0.05, β=0.2, population variance=6.342 and allowable difference=3 obtained a n=6 with the software Sample Size Calculator: One Sample Mean [36], which was approved by the animal use committee.
4.- Were the animals blocked by weight before random assignments to their respective groups?
Animals were blocked by weight before being randomly assigned to their respective groups using Microsoft Excel. We added a sentence.
Line 109-111. Animals were randomly placed into groups; random numbers were generated using the standard=RAND() function in Microsoft Excel.
5.- How was the dosing and protocol for diabetes induction selected (include citation).
The method was carried out accordance with the article by Soliman (2016). We added the reference.
Line 114-116. The rats were injected intraperitoneally with streptozotocin (STZ) (Sigma-Aldrich, St. Louis, MO, USA) at a dose of 60 mg/kg dissolved in a citrate buffer (0.1M pH 4.5), while the control rats were administered citrate buffer [37].
- Soliman, A.M. Potential Impact of Paracentrotus Lividus Extract on Diabetic Rat Models Induced by High Fat Diet/Streptozotocin. The Journal of Basic & Applied Zoology 2016, 77, 8–20, doi:10.1016/j.jobaz.2016.01.001
6.- The materials and methods section would benefit from more extensive referencing as per the methods and rationales used to select the procedures mentioned.
We added some references to the Materials and Methods.
Line 489-502.
- Soliman, A.M. Potential Impact of Paracentrotus Lividus Extract on Diabetic Rat Models Induced by High Fat Diet/Streptozotocin. The Journal of Basic & Applied Zoology 2016, 77, 8–20, doi:10.1016/j.jobaz.2016.01.001
- Buñag, R.D. Validation in Awake Rats of a Tail-Cuff Method for Measuring Systolic Pressure. J Appl Physiol 1973, 34, 279–282, doi:10.1152/jappl.1973.34.2.279.
- Chomczynski, P.; Sacchi, N. Single-Step Method of RNA Isolation by Acid Guanidinium Thiocyanate-Phenol-Chloroform Extraction. Anal Biochem 1987, 162, 156–159, doi:10.1016/0003-2697(87)90021-2.
- Bustin, S. Quantification of MRNA Using Real-Time Reverse Transcription PCR (RT-PCR): Trends and Problems. J Mol Endocrinol 2002, 29, 23–39, doi:10.1677/jme.0.0290023.
- Livak, K.J.; Schmittgen, T.D. Analysis of Relative Gene Expression Data Using Real-Time Quantitative PCR and the 2−ΔΔCT Method. Methods 2001, 25, 402–408, doi:10.1006/meth.2001.1262.
- Wu, W. A Novel Approach for Evaluating the Efficiency of SiRNAs on Protein Levels in Cultured Cells. Nucleic Acids Res 2004, 32, 17e–117, doi:10.1093/nar/gnh010.
- https://www.invivogen.com/sirna-wizard SiRNA WizardTM Online Tool.
- Multiple Sequence Alignment. https://www.ebi.ac.uk/Tools/msa/clustalo/ Clustal Omega.
7.- It is important to discuss briefly any potential (predicted) side effects especially in thinking long-term effects of such an approach like the proposed one if it were to be conducted in an extending time line experiment and what those implications may be at the clinical level.
We added information about adverse reactions with the use of siRNAs.
Line 366-369. “…but we can discard some adverse reactions that are present in all siRNAs [86]. However, the unwanted effects present in siRNAs are associated with the length of the sequences where 27 mers are more toxic than 21 mers and the vehicle and duration of the treatments can produce an immune response [87].”
8.- Similar improvements in vasculature have been reported through dietary interventions especially with rich in antioxidant fruits. It would be worth including a brief discussion in this regard to link the topic to the clinical application
These are interesting articles; for that reason, we added information mainly in the introduction about some fruits that, through antioxidants, could help at the vascular level.
Line 58-59. “…some benefits of blueberry consumption [15,16]
Comments on the Quality of English Language
We requested an English revision.
The article was revised in English (Proof-Reading-Service.com Certificate_202310-456855).
Reviewer 2 Report
Comments and Suggestions for Authors
The manuscript by Garcia-Rubio et al intends to demonstrate that inhibition of PDE5 using siRNA can restore endothelial dysfunction in streptozotocin (STZ) induced type 1 diabetes in rats. The authors show that the mRNA level of PDE5 is induced in established Type 1 diabetes rat model and injection of siRNA in jugular vein blunts the expression of PDE5. Overall the manuscript needs substantial modification and requires rewriting. The conclusion of the study is based on weak results and needs further evidence to clearly vindicate the proposed hypothesis. Following are my comments.
1) In addition to the mRNA expression of PDE5, the authors should also demonstrate the protein expression of PDE5 in Control (non-diabetic rats) and upon induction of diabetes and post siRNA-PDE5 treatment.
2) Evidences to show endothelial dysfunction is lacking and requires additional data. Protein expression of eNOS, VCAM, VE-cadherin and VASP.
3) Fig-5 – Treatment with Vehicle alone shows a significant decrease in mRNA expression of PDE5.
4) I also strongly recommend to use a positive control group in the study such as treatment with PDE5 inhibitor sildenafil to complement the data obtained using siRNA-PDE5 silencing.
Comments on the Quality of English LanguageThe English language needs improvement. it is very hard to understand the intention of the message conveyed by the authors due to wrong sentence formation.
Example : in Abstract " For this reason, the aim of this study was to design, synthesize and evaluate siRNAS targeting PDE5 to re-duce endothelial dysfunction in diabetes." This sentence conveys completely opposite meaning to the central hypothesis of the manuscript.
There are several grammar mistakes that needs attention.
Author Response
Dear reviewer;
We made all the suggested changes and tried to answer all your questions.
1.- In addition to the mRNA expression of PDE5, the authors should also demonstrate the protein expression of PDE5 in Control (non-diabetic rats) and upon induction of diabetes and post siRNA-PDE5 treatment.
This is a good recommendation, but WB was not performed in the studies because the aim was to evaluate whether an siRNA directed at PDE5A can restore endothelial dysfunction associated with diabetes, finding that it can effectively reduce it. The degradation of the mRNA PDE5A was verified by RT-PCR which is a valid technique for this purpose (Wu 2004). This effect has been reported since the discovery of siRNAs (Fire et al., 1998). Interestingly, studies with siRNAs targeting PDE5 in erectile dysfunction and endothelial cells found that PDE5 mRNA decreased, but this was also associated with a decrease in PDE5 protein expression and an increase in cGMP levels, which are linked to the nitric oxide pathway (Lin et al., 2005; Zhu et al., 2009). For this reason, by determining the decrease in PDE5 mRNA by RT-PCR, we can verify the silencing and efficacy of the treatment through functional evaluation. We added information about the method used.
Line 161-163. This method is the most sensible assay to demonstrate the siRNA efficiency because it avoids problems with the half-life of the proteins that could mask the silencing efficiency [42].
Bing Zhu, Li Zhang, Mikhail Alexeyev, Diego F Alvarez, Samuel J Strada, Troy Stevens. Type 5 phosphodiesterase expression is a critical determinant of the endothelial cell angiogenic phenotype. Am J Physiol Lung Cell Mol Physiol . 2009 Feb;296(2):L220-8. doi: 10.1152/ajplung.90474.2008. Epub 2008 Nov 21.
Wu, W. (2004). A novel approach for evaluating the efficiency of siRNAs on protein levels in cultured cells. Nucleic Acids Research, 32(2), 17e–117. https://doi.org/10.1093/nar/gnh010
Guiting Lin 1, Narihiko Hayashi, Rafael Carrion, Lung-Ji Chang, Tom F Lue, Ching-Shwun Lin. Improving erectile function by silencing phosphodiesterase-5. J Urol . 2005 Sep;174(3):1142-8. doi: 10.1097/01.ju.0000168615.37949.45.
A Fire 1, S Xu, M K Montgomery, S A Kostas, S E Driver, C C Mello. Potent and specific genetic interference by double-stranded RNA in Caenorhabditis elegans. Nature. 1998 Feb 19;391(6669):806-11. doi: 10.1038/35888.
2.- Evidences to show endothelial dysfunction is lacking and requires additional data. Protein expression of eNOS, VCAM, VE-cadherin and VASP.
We used the definition of endothelial dysfunction to relate the results. We added a sentence.
Line 322-326. Endothelial dysfunction is characterised by a reduction in nitric oxide generation or an unbalanced distribution of the relaxing and contracting pressures produced by endothelial cells [62] and has been defined as an attenuated response of the blood vessels to endothelium-dependent vasodilators such as acetylcholine (ACh) and bradykinin [63].
3.- Fig-5 – Treatment with Vehicle alone shows a significant decrease in mRNA expression of PDE5.
That is a common effect with all of the transfection vehicles; for that reason, we added a sentence.
Line 371-374. The decrease in PDE5 expression due to the vehicle could be due to the cytotoxic effects that different transfection agents usually induce [87,88]; however, the effect of siRNAs is significantly different from the effects of the vehicle, suggesting that the effect is due to silencing, as demonstrated in the determinations of mRNA.
4.- I also strongly recommend to use a positive control group in the study such as treatment with PDE5 inhibitor sildenafil to complement the data obtained using siRNA-PDE5 silencing.
That recommendation is very important, and we do not discount the use of a positive control in the future, but we want only to show beneficial effects of the PDE5A-siRNA at the present time, like the Lin study, which showed that PDE5i is not used, even in erectile dysfunction. Also, we have to consider that none of the PDE5Ai are genuinely selective for PDE5A. Most of the adverse reactions are due to its non-selective interaction with other PDE isoenzymes; for example, blue colour vision, back pain, myalgia, dyspepsia, headache, flushing, hypotension and dizziness may be primarily due to the interactions with PDE6 and PDE11 (Ballard et al., 1998; Bischoff, 2004; B. P. Smith & Babos, 2022).
Guiting Lin 1, Narihiko Hayashi, Rafael Carrion, Lung-Ji Chang, Tom F Lue, Ching-Shwun Lin. Improving erectile function by silencing phosphodiesterase-5. J Urol . 2005 Sep;174(3):1142-8. doi: 10.1097/01.ju.0000168615.37949.45.
Ballard, S. A., Gingell, C. J., Tang, K., Turner, L. A., Price, M. E., & Naylor, A. M. (1998). Effects of sildenafil on the relaxation of human corpus cavernosum tissue in vitro and on the activities of cyclic nucleotide phosphodiesterase isozymes. The Journal of Urology, 159(6), 2164–2171.
Bischoff, E. (2004). Potency, selectivity, and consequences of nonselectivity of PDE inhibition. International Journal of Impotence Research, 16(S1), S11–S14. https://doi.org/10.1038/sj.ijir.3901208.
Smith, B. P., & Babos, M. (2022). Sildenafil. StatPearls Publishing
Comments on the Quality of English Language
The article was revised in English (Proof-Reading-Service.com Certificate_202310-456855)
We rephrased the sentence.
Line 23-24. “For this reason, the aim of this study was to evaluate whether siRNAS targeting PDE5A can reduce the endothelial dysfunction associated with diabetes.”
Reviewer 3 Report
Comments and Suggestions for Authors
In this study, the authors investigated the effect of siRNA-mediated downregulation of PDE5 in rescuing the vascular in a STZ-induced type1 diabetes rat model. The study is interesting but contains several critical caveats that render it inappropriate for acceptance as per the journal standard in its present form. Please consider the following suggestions to revise and re-submit the manuscript in the future.
1) Since STZ induction can induce both type1 and type2 in a dose-dependent manner, it's important to cite any relevant reference(s) to confirm that the dose used was appropriate for developing type 1 symptoms.
2) Please provide western blot results showing the specificity of the custom-designed siPDE5A in both rat and human cell models.
3) Although the authors mentioned the relationship of PDE5 with cGMP and NO signaling pathways, however, there was no relevant data to show the effect of PDE5A downregulation on these pathways. The authors may consider restructuring these sections to include only study-relevant topics.
4) It's not clear from the overall results what is the exact role of PDE5A, mechanistically, in type1 diabetes.
Comments on the Quality of English Language
The whole manuscript contains lots of typos, grammatical errors, and poor-quality English language. This needs to be thoroughly revised and rewritten to meet the publication standard of the journal.
Author Response
Dear reviewer;
We made all the suggested changes and tried to answer all your questions.
1.- Since STZ induction can induce both type1 and type2 in a dose-dependent manner, it's important to cite any relevant reference(s) to confirm that the dose used was appropriate for developing type 1 symptoms
We added a sentence and a reference about the selected dose.
Line 116. “….administered citrate buffer [37].”
Line 120-122. “..blood samples were obtained from the rat tail by a small cut and glucose levels greater than ≥300 mg/dL were considered type 1 diabetes [37]”
- Soliman, A.M. Potential Impact of Paracentrotus Lividus Extract on Diabetic Rat Models Induced by High Fat Diet/Streptozotocin. The Journal of Basic & Applied Zoology 2016, 77, 8–20, doi:10.1016/j.jobaz.2016.01.001.
2.- Please provide western blot results showing the specificity of the custom-designed siPDE5A in both rat and human cell models.
In the studies, WB was not performed, because the aim was to evaluate whether an siRNA directed at PDE5A can restore the endothelial dysfunction associated with diabetes, showing that it can effectively reduce it. The specificity of this siRNA PDE5A was verified through the alignment with human and rat sequences. Furthermore, the degradation of PDE5A mRNA was verified by RT-PCR, which is a valid technique for this purpose (Wu 2004). This effect has been reported since the discovery of siRNAs (Fire et al., 1998). Interestingly, studies with siRNAs targeting PDE5 in erectile dysfunction and endothelial cells found that PDE5 mRNA levels decreased, but this was also associated with a decrease in PDE5 protein expression and an increase in cGMP levels, showing that it is connected to the nitric oxide pathway (Lin et al., 2005; Zhu et al., 2009). For this reason, by determining the decrease of PDE5 mRNA by RT-PCR, we can verify the silencing and the efficacy of the treatment through functional evaluation. We added information about the method used:
Line 161-163. This method is the most sensible assay to demonstrate the siRNA efficiency because it avoids problems with the half-life of the proteins that could mask the silencing efficiency [42].
Bing Zhu, Li Zhang, Mikhail Alexeyev, Diego F Alvarez, Samuel J Strada, Troy Stevens. Type 5 phosphodiesterase expression is a critical determinant of the endothelial cell angiogenic phenotype. Am J Physiol Lung Cell Mol Physiol . 2009 Feb;296(2):L220-8. doi: 10.1152/ajplung.90474.2008. Epub 2008 Nov 21.
Wu, W. (2004). A novel approach for evaluating the efficiency of siRNAs on protein levels in cultured cells. Nucleic Acids Research, 32(2), 17e–117. https://doi.org/10.1093/nar/gnh010
Guiting Lin 1, Narihiko Hayashi, Rafael Carrion, Lung-Ji Chang, Tom F Lue, Ching-Shwun Lin. Improving erectile function by silencing phosphodiesterase-5. J Urol . 2005 Sep;174(3):1142-8. doi: 10.1097/01.ju.0000168615.37949.45.
A Fire 1, S Xu, M K Montgomery, S A Kostas, S E Driver, C C Mello. Potent and specific genetic interference by double-stranded RNA in Caenorhabditis elegans. Nature. 1998 Feb 19;391(6669):806-11. doi: 10.1038/35888.
3.- Although the authors mentioned the relationship of PDE5 with cGMP and NO signaling pathways, however, there was no relevant data to show the effect of PDE5A downregulation on these pathways. The authors may consider restructuring these sections to include only study-relevant topics.
A sentence was added.
Line 358-363. For this reason, we cannot exclude the possibility that the mechanisms of action are shared with respect to the nitric oxide pathway; also, some studies showed that a decrease in mRNA with the siRNA of PDE5A causes a decrease in the protein, which is related to an increase in cGMP, suggesting a mechanism by which the siRNA targets PDE5 in the nitric oxide pathway [81,84].
4.- It's not clear from the overall results what is the exact role of PDE5A, mechanistically, in type1 diabetes.
A decrease in vasodilatory responses has been reported in diabetes, as well as an increase in the expression of PDE5. The silencing of this enzyme partially reverses this decrease, probably by increasing cGMP, which has been reported with the use of siRNA PDE5.
Studies with siRNAs targeting PDE5 in erectile dysfunction and endothelial cells found that PDE5 mRNA levels decreased, but this was also associated with a decrease in PDE5 protein expression and an increase in cGMP levels, showing that it is connected to the nitric oxide pathway (Lin et al., 2005; Zhu et al., 2009).
A paragraph was added.
Line 358-363. For this reason, we cannot exclude the possibility that the mechanisms of action are shared with respect to the nitric oxide pathway; also, some studies showed that a decrease in mRNA with the siRNA of PDE5A causes a decrease in the protein, which is related to an increase in cGMP, suggesting a mechanism by which the siRNA targets PDE5 in the nitric oxide pathway [81,84].
Comments on the Quality of English Language
The article was revised in English (Proof-Reading-Service.com Certificate_202310-456855).
Round 2
Reviewer 1 Report
Comments and Suggestions for Authors
The authors have addressed reasonably the reviewer's points.
Reviewer 3 Report
Comments and Suggestions for Authors
Although there are some language errors and typos, the authors were able to clarify the points raised. So the manuscript is acceptable for publication.
Comments on the Quality of English LanguageSome English language editing and grammatical corrections are necessary in the final version.